# Adversarial Policy Gradient for Learning Graph-Based Representation in Human Visual Processing

**Subhrasankar Chatterjee, Subrata Pain & Debasis Samanta**
Department of Computer Science
Indian Institute of Technology
Kharagpur, West Bengal 721302, India

## Abstract

This article discusses the challenges in modeling the neural mechanisms underlying human visual processing and the use of graph-based representations to capture inter-region relationships in visual processing. While graphs have shown promise in analyzing neural responses, learning an optimal graph representation from limited data is challenging, as there is no ground truth to learn from. To address this, the authors propose a novel approach to graph-based representation using Adversarial Policy Gradient, which involves an adversarial game between the Policy Network and the Reward Network to generate a better graph representation.

## 1 Introduction

Understanding the fundamental differences in performance between human visual processing and artificially intelligent systems (AI) has been a subject of significant interest in neuroscience and AI (Kamitani & Tong, 2005; Du et al., 2018; Palazzo et al., 2020). While computer vision models typically generate latent representation spaces that are then mapped to specific tasks (Robert, 2019), the neural mechanisms underlying human visual processing remain an area that requires further exploration. Neuroscience and AI researchers have utilized computational modeling techniques, such as functional Magnetic Resonance Imaging (fMRI), to investigate the human visual system during visual tasks (Haynes & Rees, 2005; Thirion et al., 2007). Encoding models estimate neural responses from stimuli while decoding models predict stimuli from neural responses (Naselaris et al., 2011; Kriegeskorte, 2011; Haxby, 2012; Kay, 2018). These models often rely on handcrafted or learnable computer vision models (Wu et al., 2006; Kay et al., 2008; Agrawal et al., 2014; Güçlü & van Gerven, 2014; Wen et al., 2018a; Han et al., 2019; Wen et al., 2018b; Cui et al., 2021) to capture the stimulus-response relationship.

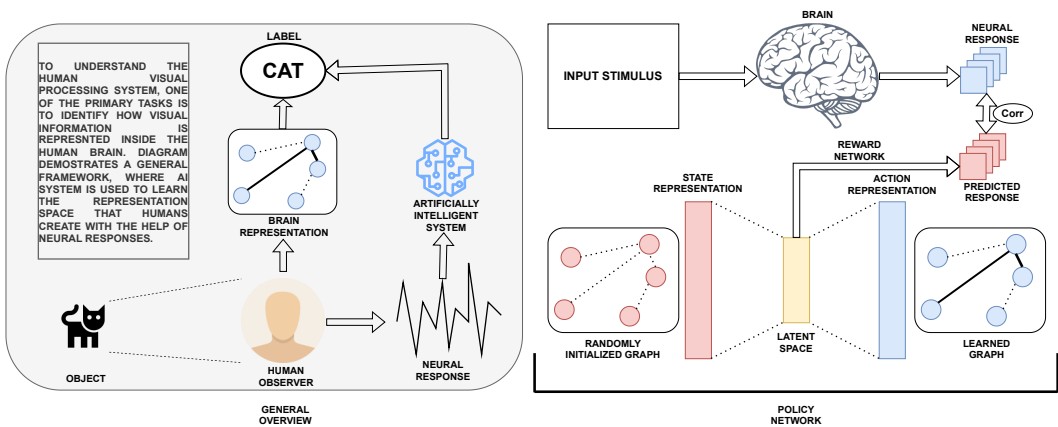

Figure 1: The General Architecture of our proposed methodology.

However, the representation space for human visual processing, particularly the inter-region relationships, requires more exploration. Graph-based representations have shown promise in capturing the characteristics of neural responses (Conturo et al., 1999; Dipasquale et al., 2017; Mohanty et al., 2020; Gilson et al., 2019; Deshpande & Wang, 2022). They have been used for decoding neural data, identifying neurological biomarkers (Li et al., 2021), and improving visual decoding accuracy (Li et al., 2022; Meng & Ge, 2022). Nevertheless, learning an optimal graph representation from limited data without ground truth remains challenging. This paper proposes an Adversarial Policy Gradient approach for graph-based representation of human visual processing. Our approach aims to generate an optimal graph representation that captures the inter-region relationships in visual processing by training a Policy Network and a Reward Network using an adversarial game. This paper presents the methodology, training procedure, and evaluation of the proposed approach and discusses its potential applications in understanding the mechanisms of human visual processing.

## 2 METHODOLOGY

The proposed methodology is shown in Figure 1. A Graph $\mathcal{G}$ is represented by a vector $S_t$, obtained by flattening the upper triangular part of the adjacency matrix of $\mathcal{G}$. We begin with a random graph $\mathcal{G}_0$ ($V_0$, $E_0$), where $E_0$ is the initial edges. We seek to find a graph $\mathcal{G}_f$ with final edges $E_f$, represented by the vector $S_f$, that accurately captures the behavior of inter-ROI neural responses. A vanilla neural network called the Policy Network, parameterized by random weights $\theta$, takes in the state representation $S_t$ of the graph $\mathcal{G}_t$ at instance $t$ and outputs a new state representation $S_{t+1}$, considered an improved representation $G_{t+1}$. Policy Gradient algorithm was used to train the Policy Network. The Reward $R$ is defined as the correlation between the predicted neural response from the state $S_{t+1}$ and the original response. The Reward $R_t$ at instance $t$ is computed by another learnable neural network called the Reward Network. However, the Reward Network cannot be trained using supervised learning with the original responses as the target label since with every new instance $t + n$, the input state to the Reward Network changes to $S_{t+n}$. We address the problem with Adversarial training, where the Policy Network and the Reward Network play an adversarial game. The entire framework can be formulated as a minimax optimization problem to maximize the Reward and minimize the loss of the predicted neural response.

## 3 RESULTS

The proposed Adversarial Policy Gradient approach was implemented using PyTorch. Dummy data was used for training and testing the model. The model achieved a test accuracy of 62% for the image classification problem on the Kay datasetKay et al. (2008) and a mean squared error of 0.1 on the test data.

## 4 DISCUSSION

The Adversarial Policy Gradient algorithm provides a novel approach for learning an optimal graph representation of human visual processing. The model captures the inter-region relationships in visual processing by training a Policy Network and a Reward Network in an adversarial game. The proposed approach addresses the challenge of limited labeled data availability and achieves promising results on the Kay 2008 dataset.

## 5 CONCLUSION

This paper presents an Adversarial Policy Gradient approach for graph-based representation of human visual processing. The proposed approach addresses the challenge of limited labeled data availability and learns an optimal graph representation by training a Policy Network and a Reward Network in an adversarial game. The results on Kay dataset demonstrate the effectiveness of the proposed approach. Further research is needed to apply the approach to real-world visual processing data and investigate its performance compared to existing models.

URM STATEMENT

The authors acknowledge that at least one key author of this work meets the URM criteria of ICLR 2023 Tiny Papers Track.

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

## A  ADVERSARIAL POLICY GRADIENT

The Modified Policy Gradient algorithm is used to update the Policy Network by maximizing the expected reward. The Policy Network generates an updated state representation given the current state representation, which is used to generate the graph. Let $\theta$ be the parameters of the Policy Network, $s_t$ be the current state representation and $s_{t+1}$ be the next state representation. The Policy Network can be represented as a function $\pi(s_{t+1}|s_t, \theta)$, where $s_{t+1}$ is the next state derived at time $t$ given state $s_t$.

The reward is generated by a separate neural network called the Reward Network, which predicts the neural response based on the current graph representation. Let $W$ be the parameters of the Reward Network, and $r_t$ be the reward at time $t$. The Reward Network can be represented as a function $R(s_t, W)$.

The goal is to maximize the expected reward by updating the Policy Network parameters $\theta$. The update rule can be written as:

$$\Delta\theta = \alpha\nabla_\theta \log \pi(s_{t+1}|s_t,\theta)r_t$$

where $\alpha$ is the learning rate and $\nabla_\theta$ is the gradient with respect to $\theta$. The gradient of the expected reward with respect to $\theta$ is calculated using the Policy Gradient theorem:

$$\nabla_\theta J(\theta) = E_\pi[\nabla_\theta \log(s_{t+1}|s_t,\theta)r_t]$$

where $J(\theta)$ is the objective function to be maximized. The expectation is taken over all possible current state-next state pairs under the policy $\pi$.

To update the Reward Network, we use the mean squared error loss between the predicted neural response and the actual neural response as the loss function. Let $y_t$ be the actual neural response and $y_hat_t$ be the predicted neural response. The loss function can be written as:

$$L(W) = (y_t - \hat{y}_t)^2$$

The parameters of the Reward Network, $W$, are updated using stochastic gradient descent by minimizing the loss function:

$$\Delta W = -\alpha\nabla_W L(W)$$

where $\alpha$ is the learning rate and $\nabla_W$ is the gradient with respect to $W$.

The minimax optimization problem for the proposed Adversarial Policy Gradient approach can be formulated as follows:

$$min_W max_\theta Z(W,\theta)$$

where $\theta$ represents the parameters of the Policy Network, $W$ represents the parameters of the Reward Network, and $Z(W,\theta)$ is the objective function defined as:

$$Z(W,\theta) = E_\tau[\sum_{t=1}^{T} R(s_t, s_{t+1}) - \beta KL(\pi_\theta||\pi_W|s_t)]$$

where $\tau$ represents a trajectory of states and actions, $\beta$ is a hyperparameter that controls the strength of the KL divergence regularization term, $KL(\pi_\theta||\pi_W|s_t)$ represents the KL divergence between the Policy Network and the Reward Network's policies, and $\pi_\theta$ and $\pi_W$ represent the probability distributions over actions or states, parameterized by $\theta$ and $W$, respectively.

Overall, the Adversarial Policy Gradient algorithm involves an adversarial game between the Policy Network and the Reward Network, where the Policy Network generates an updated state representation to improve the graph representation, and the Reward Network generates a reward based on the current graph representation. The Policy Network is updated using the Policy Gradient algorithm with the reward generated from the Reward Network. The adversarial game continues until the graph representation adequately captures the behavior of inter-ROI neural responses.

## B  METHODOLOGICAL FRAMEWORK

### B.1  PROBLEM FORMULATION

The problem is to learn an optimal graph representation for human visual processing without access to labeled data. Given the limited availability of labeled data, supervised learning algorithms cannot be directly applied. The objective is to generate a graph representation that captures the inter-region relationships in visual processing by training a Policy Network and a Reward Network using the Adversarial Policy Gradient algorithm.

## B.2 ADVERSARIAL POLICY GRADIENT

The Adversarial Policy Gradient algorithm involves an adversarial game between the Policy Network and the Reward Network. The Policy Network generates an updated state representation to improve the graph representation, while the Reward Network generates a reward based on the current graph representation. The Policy Network is updated using the Policy Gradient algorithm with the reward generated from the Reward Network. The adversarial game continues until the graph representation adequately captures the behavior of inter-ROI neural responses.

## B.3 MODEL ARCHITECTURE

The Policy Network and the Reward Network are implemented as neural networks. The Policy Network takes the current state representation and the graph structure as inputs and generates an updated state representation. The Reward Network takes the current graph representation as input and predicts the reward. The neural networks can be implemented using deep learning frameworks such as PyTorch.

## B.4 TRAINING PROCEDURE

The training procedure involves iteratively updating the Policy Network and the Reward Network using the Adversarial Policy Gradient algorithm. The Policy Network is updated using the Policy Gradient algorithm, where the gradient of the expected reward concerning the Policy Network parameters is calculated. The Reward Network is updated using stochastic gradient descent, where the mean squared error loss between the predicted reward and the actual reward is minimized. The learning rate was set to 0.01. The batch size was 32, and the vector size representing the inter-region graph was 21x1. The number of training episodes was set to 100.

## B.5 EVALUATION

The trained model is evaluated using test data. The test data, consisting of states and actions, is fed into the trained model, and the policy probabilities and predicted rewards are obtained. The model's performance can be evaluated using various metrics, such as accuracy and mean squared error.

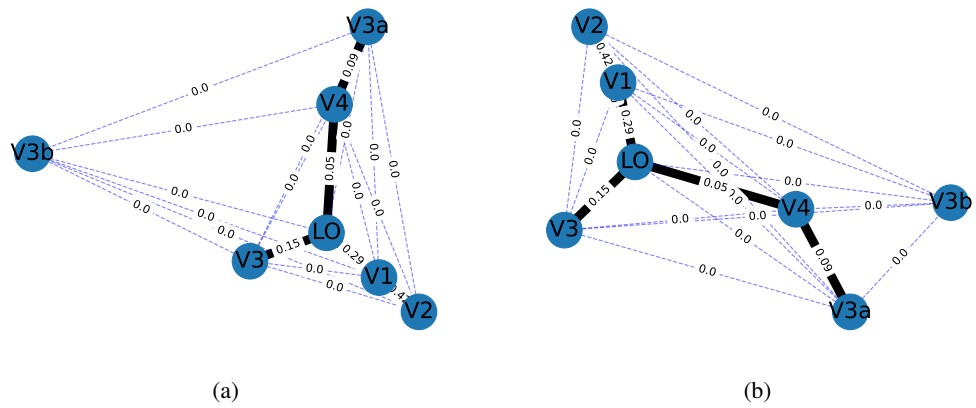

(a)                                        (b)

Figure 2: Graphical Representation of a fMRI data of an image.

