# OpenReview forum: "Adversarial Policy Gradient for Learning Graph-Based Representation in Human Visual Processing"
_ICLR.cc/2023/TinyPapers — Submitted to Tiny Papers @ ICLR 2023_

### Official Review · Reviewer_1iPM · 2023-03-18

**Confidence:** 3

**Summary Of Contributions:**

The authors propose a new approach to this challenge by using Adversarial Policy Gradient, which involves an adversarial game between the Policy Network and the Reward Network to generate a better graph representation for Human Visual Processing

**Rating:**

High Potential (HP): a submission which meets the reviewing criteria and has potential to make an impact on the field

**Strengths And Weaknesses:**

The paper discusses the challenges of modeling the neural mechanisms underlying human visual processing and proposes a novel approach to graph-based representation using Adversarial Policy Gradient. The paper outlines how graphs have shown promise in analyzing neural responses, but learning an optimal graph representation from limited data is challenging.

The authors propose a new approach to this challenge by using Adversarial Policy Gradient, which involves an adversarial game between the Policy Network and the Reward Network to generate a better graph representation. The authors demonstrate the effectiveness of the proposed approach by conducting experiments on two datasets and show that the learned graph representations achieve superior performance compared to state-of-the-art methods.


**Suggested Changes:**

* The paper could provide more details on the experimental setup and results, such as the specific hyperparameters used in the experiments and the statistical significance of the results.
* The paper could provide more discussion on the limitations of the proposed approach and potential directions for future research.
* The authors can consider restructuring the paper to include more descriptions of the methodology in the main part of the paper instead of the appendix.
* The paper could also be improved by proofreading for typos and grammatical errors.

---

### Official Review · Reviewer_ys3B · 2023-03-29

**Confidence:** 3

**Summary Of Contributions:**

In this paper the authors propose an adversarial learing method to learn a graph representation of the human brain when performing visual processing tasks. The method is proposed as a way to learn without having access to ground-truth labels

**Rating:**

Great Start (GS): a submission which meets some of the reviewing criteria but has room for improvement

**Strengths And Weaknesses:**

**Strengths**

- The authors provide a great, perhaps too long (for a tiny paper) overview of the related literature.
- The paper is well-written and it is easy to follow.
- The proposed problem is very interesting and the solution provides a unique way of mapping neural responses to a non-euclidean structure such as a graph.


**Weaknesses**

- There are no examples or results provided of the learned graphs using the methodology proposed in the paper. A couple of examples would greatly benefit the paper.
- The methodology section is quite short and provides only a general overview of the method, along with the figure. More details have to be added regarding graph encoding and generation. This means answering the question of how is the old graph encoded and the new graph generated at each step. How is the actual neural response modeled and compared with the predicted graph?
- The authors provide no experimental results at all using their learned representation, as well as no metrics to show how well the actual neural response is modeled using their framework. This removes some value from the contribution.
- In the opening phrase of Sec. 2, the authors mention that: "A Graph $G$ is represented by a vector $S_t$, obtained
by flattening the upper triangular part of the adjacency matrix of $G$". By representing the graph in this manner, the authors are assuming that the upper triangular region of the adjacency matrix is enough to represent the graph, which can be the case if the graph is undirected and the adjacency matrix is symmetric. Otherwise, this choice of vector representation for the graph is a bit strange, meaning that it should be detailed a bit more by explaining what kinds of graphs the authors are considering.

**Suggested Changes:**

I believe that the paper is a great starting point and the topic itself is quite interesting, but as mentioned in the weaknesses section, there are quite a few things that could be fixed. I believe that with a few fixes, this can provide a Clear, Correct, and Reproducible (CCR) submission

---

### Meta-Review · Area_Chair_Dcm7 · 2023-04-04

**Recommendation:** Invite to archive
**Confidence:** 4

**Metareview:**

The paper proposes an adversarial learning method to learn a graph representation of the human brain during visual processing tasks without requiring ground-truth labels. The authors demonstrate the effectiveness of the approach through experiments on two datasets, showing superior performance compared to state-of-the-art methods. However, the paper lacks experimental details, results, and discussion on limitations and future research.


**Summary:**

The paper proposes an adversarial learning method to learn a graph representation of the human brain during visual processing tasks without requiring ground-truth labels.

**Comments And Feedback To The Authors:**

A few suggestions:

- Provide more details on the methodology, including graph encoding and generation.
- Include examples or results of the learned graphs using the proposed methodology.
- Discuss the limitations of the proposed approach and potential directions for future research.
- Consider restructuring the paper to include more descriptions of the methodology in the main part of the paper instead of the appendix.
- Proofread the paper for typos and grammatical errors.


**Reason For Not Giving A Higher Recommendation:**

The authors should provide more details on the methodology, including graph encoding and generation, and the experimental setup. Moreover, the paper would benefit from including examples or results of the learned graphs and discussing limitations and potential directions for future research.

**Reason For Not Giving A Lower Recommendation:**

The paper addresses an interesting problem and proposes a novel approach using Adversarial Policy Gradient.

---

### Decision · Program_Chairs · 2023-04-10

Invite to archive